# Characteristics and Surface Serviceability for Cryogenic Milling Mg-1.6Ca-2.0Zn Medical Magnesium Alloy

**DOI:** 10.3390/jfb13040179

**Published:** 2022-10-07

**Authors:** Xuan Guo, Guodong Liu, Shunheng Sang, Qichao Lin, Yang Qiao

**Affiliations:** 1School of Mechanical Engineering, University of Jinan, Jinan 250022, China; 2Shandong Provincial Measurement and Testing Center, Jinan 252399, China

**Keywords:** magnesium alloy, cryogenic milling, tensile strength, service performance, liquid nitrogen cooling

## Abstract

Magnesium alloy has great potential as a new medical metal material because of its good biocompatibility and biodegradability. However, because of the active chemical properties of magnesium alloy, it is easy to react with oxygen and cutting fluid to release hydrogen. In this paper, by cutting magnesium alloys prepared under different cooling conditions, the phase composition of the machined surface was studied. Tensile strength and elongation were studied through tensile experiments at different temperatures. The effect of cryogenic milling on the service performance of a magnesium alloy machined surface was studied by the friction and wear test and electrochemical corrosion test. The results show that cryogenic milling contributes to the formation of the second phase of magnesium alloy, which has the effect of corrosion resistance, and has better tensile strength and elongation. Through the friction and wear test, it is found that the average friction coefficient decreases by about 7.4%, and the wear amount decreases by about 10% in the liquid nitrogen cooling environment. Through the electrochemical corrosion test, it was found that the oxide film formed in the liquid nitrogen cooling environment was more compact and uniform, and the crystal refinement of the surface layer was better.

## 1. Introduction

Medical metal materials play an important role in the repair and replacement of bone defects caused by osteoarthrosis and trauma [1]. Magnesium is one of the main cations in human cells, and it is a necessary substance for basic biochemical reactions of various cells in the body [2]. In recent years, alloying has been a widely used method to improve the degradation rate of pure magnesium [3], and the addition of calcium and zinc can improve the corrosion resistance by improving the mechanical properties [4,5,6,7]. In the field of orthopedic medicine, compared with traditional bone plate materials, such as titanium alloy, the elastic modulus of magnesium alloy is closer to that of the human bone, which can effectively slow down the “stress shielding” effect [8,9,10,11,12,13]. In addition, magnesium alloy can be degraded in the human body due to its good biochemical compatibility and unique biodegradability, which avoids the second removal surgery [14]. However, due to the active chemical properties of magnesium, the cutting process will produce heat release, which accelerates the exothermic reaction between chips and oxygen in the air. This condition will burn the machined surface [15] and cause deformation of the workpiece [16,17]. Therefore, the correct use of cutting fluid can reduce the occurrence of such problems. Compared with conventional cutting fluid, it is easy to cause corrosion of the machined surface [18], and cryogenic milling with liquid nitrogen can not only avoid corrosion, but also meet the requirements of “green manufacturing” [19,20,21,22], so it has become a research hotspot in the cross-field of medicine and industry.

Cryogenic milling with liquid nitrogen can reduce cutting temperature, prolong tool life, and improve surface service performance. This is because when liquid nitrogen is used as the cooling medium, liquid nitrogen can be sprayed to the contact area of the cutting chips. This method can reduce the regional temperature, enhance the brittleness of the material, and effectively prevent the deformation and chip burning of magnesium alloy, thereby facilitating cutting, enhancing corrosion resistance, improving the quality of the machined surface, and prolonging the service life of magnesium alloy workpieces [23,24]. Yan [25] et al. compared experiments with magnesium alloy with cold air under pressure or dry cutting, and found that the former milling improved the surface quality and prevented magnesium alloy from burning easily during dry cutting. Desai [26] et al. studied the influence of processing parameters on the properties of medical magnesium alloys. The results show that increasing cutting speed can improve the corrosion resistance and effectively alleviate pitting corrosion.

In this paper, calcium and zinc elements were added into pure magnesium alloy [27], and the tensile test was carried out to determine the mechanical properties of the material, such as tensile strength, yield strength, and elongation. In this experiment, the phase composition of a magnesium alloy machined surface under different cooling conditions was studied by vertical milling test. The effect of low-temperature cutting on the service performance of the magnesium alloy machined surface was studied by the friction and wear test and electrochemical corrosion test. This can provide some reference for the preparation and cutting of high-performance magnesium alloys.

## 2. Materials and Methods

### 2.1. Preparation of Test Materials

Among the magnesium alloys prepared by Zhang Erlin [28,29] and others, it was found that 2%Zn showed good corrosion resistance, and Wang [30] and others prepared magnesium alloys by powder metallurgy. Their research found that Mg-1.6%Ca-2.0%Zn alloy had the best corrosion resistance and mechanical properties. Therefore, based on previous studies, Mg-Ca-Zn alloy was prepared in this experiment, in which the calcium content was 1.6% and the zinc content was 2.0%. An X-ray energy dispersive spectrometer (EDS) was used to test the element content of the prepared magnesium alloy, and the results met the expectations, as shown in Table 1.

### 2.2. Magnesium Alloy Cutting

Magnesium alloy is cut by milling, and the machining center model is the YCM-V116B vertical milling machine, and the specific parameters of the machining center are shown in Table 2. The selected tool is the integral cemented carbide keyway milling cutter produced by Zhuzhou Diamond Cutting Tools Co., Ltd. (Zhuzhou, China), model GM-2E-D20.0, and the keyway milling cutter parameters are shown in Table 3. The purpose of this experiment is to explore the phase composition of the machined surface of magnesium alloy under different cooling conditions. The cooling methods include dry cutting, oil-based cutting fluid, water-based cutting fluid, and cryogenic milling. Other parameters are consistent. The cutting speed *v_c_* is 180 m/min, the axial cutting depth *a_p_* is 10 mm, the radial cutting depth *a_e_* is 1.8 mm, and the feed rate *f_z_* is 0.1 mm/Z.

After the machining is finished, the test piece is cut into 15 mm × 10 mm × 6 mm test pieces by using the electric spark numerical control cutting machine, so that it can be conveniently fixed on the workbench of the friction and wear testing machine. The machine tool for cutting the test pieces is the electric spark numerical control led by a microcomputer produced by Zhengtai Numerical Control Machine Tool Co., Ltd. (Shenzhen, China). The model is DK7735, and the maximum processing range is 450 mm × 350 mm. In order to prevent the milled surface from being oxidized or corroded in the process of wire electrical discharge machining, a layer of insulating adhesive tape is pasted on the milled surface, and after the cutting is completed, the adhesive tape is removed. To prevent impurities from remaining on the milled surface, place the cut sample in the acetone for ultrasonic cleaning for 10 min. After cleaning, take it out with tweezers, and blow it dry in the same direction with a blower. To prevent impurities from sticking on the milled surface, place the sample in absolute ethanol for ultrasonic cleaning for 5 min; then take it out with tweezers and blow it dry in the same direction with a hair dryer. The finished test piece is shown in Figure 1, with a size of 35 mm × 30 mm × 15 mm, and then each test piece is marked. In order to locate the workpiece better and eliminate the interference of other factors, draw a straight line 15 mm from the bottom edge, which plays a positioning role when clamping the workpiece. An X-ray diffractometer was used to analyze the phase composition of the machined surface. The instrument model was Bruker D8 Advance, the scanning speed was 2°/min, and the scanning range was 20–90. After the test, four groups of data were exported and compared.

After the machining is finished, the test piece is cut into 15 mm × 10 mm × 6 mm test pieces by using the electric spark numerical control cutting machine, so that it can be conveniently fixed on the workbench of the friction and wear testing machine. The machine tool for cutting the test pieces is the electric spark numerical control led by a microcomputer produced by Zhengtai Numerical Control Machine Tool Co., Ltd. (Shenzhen, China). The model is DK7735, and the maximum processing range is 450mm × 350 mm. In order to prevent the milled surface from being oxidized or corroded in the process of wire electrical discharge machining, a layer of insulating adhesive tape is pasted on the milled surface, and after the cutting is completed, the adhesive tape is removed. To prevent impurities from remaining on the milled surface, place the cut sample in the acetone for ultrasonic cleaning for 10 min. After cleaning, take it out with tweezers, and blow it dry in the same direction with a blower. To prevent impurities from sticking on the milled surface, place the sample in absolute ethanol for ultrasonic cleaning for 5 min; then take it out with tweezers and blow it dry in the same direction with a hair dryer. The finished test piece is shown in Figure 1, with a size of 35 mm × 30 mm × 15 mm, and then each test piece is marked. In order to locate the workpiece better and eliminate the interference of other factors, draw a straight line 15 mm from the bottom edge, which plays a positioning role when clamping the workpiece. An X-ray diffractometer was used to analyze the phase composition of the machined surface. The instrument model was Bruker D8 Advance, the scanning speed was 2°/min, and the scanning range was 20–90. After the test, four groups of data were exported and compared.

### 2.3. Magnesium Alloy Tensile Test

The tensile test was carried out on an electronic universal testing machine. The model of the testing machine is CMT610, and the test tensile speed is 4 mm/min. The tensile test pieces are processed by wire cutting, and is designed and processed according to the size requirements of the national standard GB/T228.3-2019 Metallic Materials–Low Temperature Test Method. The size of the test pieces are shown in Figure 2, and the gauge distance is designed to be 50 mm. As the electronic universal testing machine itself does not have the heat preservation function, it needs a heat preservation device to ensure that the test pieces keep a certain temperature during stretching, so the heat preservation device shown in Figure 3 is designed. The thermal insulation device is divided into two symmetrical parts, the inside is sponge, the outside is tinfoil, and a groove is reserved in the middle to hold the workpiece. Figure 4 shows the relative position of the specimen and the thermal insulation device.

The low-temperature tensile test was designed at seven temperatures, namely, 27 °C, 0 °C, −40 °C, −80 °C, −120 °C, −160 °C, and −196 °C, and the test was repeated three times at each temperature. Before the test, pour absolute ethanol into the cooling barrel, and then pour a small amount of liquid nitrogen to form a mixed solution. Measure the temperature of the mixed solution with a digital thermometer, and add liquid nitrogen to make the mixed solution reach the expected temperature. After that, soak the test pieces in the mixed solution for 15 min. During this period, observe the digital thermometer, and maintain the temperature of the mixed solution by continuously adding liquid nitrogen. Finally, the three test pieces are directly immersed in a liquid nitrogen tank (liquid nitrogen tank is filled with liquid nitrogen at −196 °C) for 15 min; then the test pieces are taken out, quickly stuffed into a heat preservation device, and stretched on a universal testing machine. After the test is completed, the data are exported. During the test, it was found that the mixed solution of alcohol and liquid nitrogen had solidified at −160 °C, and the test pieces could not be taken out, so the tensile test at this temperature could not be carried out. Therefore, only six groups of tensile data were actually obtained.

After the tensile test, take off the fracture of the tensile test pieces and observe its fracture morphology. The instrument used is a scanning electron microscope produced by FEI Company of the United States, and its model is Quanta 250 FEG. Then, the metallographic structure of the fracture was observed, and the fracture of the tensile test pieces was taken down for cold inlay, grinding, and polishing. The polished surface was gently wiped with a nitric acid alcohol corrosive solution with 4% solubility, then quickly washed with distilled water and dried in the same direction. The metallographic structure of the fracture of the tensile test pieces was observed using a 4XC metallographic microscope produced by Shanghai Optical Instrument No. 1 Factory.

### 2.4. Service Performance Test of Magnesium Alloy

#### 2.4.1. Friction and Wear Test of Magnesium Alloy

The friction and wear tests are carried out on an Rtec MFT-50 multifunctional friction and wear tester, which is a multifunctional friction and wear tester produced by Rtec Instruments Company of the United States. It can realize linear reciprocating and rotary friction and wear tests, and can carry out ball–disc friction and wear, pin–disc friction, and wear tests according to different friction pairs. The testing machine is equipped with a white light interferometer, which can observe the wear morphology and measure the wear volume. In this test, the test method of linear reciprocating contact wear is adopted; that is, the grinding steel ball contacts with the surface of the sample under a fixed load and performs linear reciprocating motion continuously. The GCr15 bearing steel ball with a diameter of 6.35 mm is used for the grinding steel ball; the applied load is 10 N, the frequency is 2 Hz, and the time is set to 30 min.

#### 2.4.2. Magnesium Alloy Corrosion Test

The electrochemical workstation produced by Shanghai Chenhua Co., Ltd. (Shanghai, China), was used in the electrochemical test, and the traditional three-electrode system was used in the test, which were working electrode (magnesium alloy milled surface sample), reference electrode (saturated calomel electrode), and auxiliary electrode (platinum electrode). Cut the sample into a size of 10 mm × 10 mm × 5 mm. As the corrosion resistance of the milled surface is explored, the milled surface of the sample is in contact with the electrochemical solution, and the contact area is 10 mm × 10 mm. The other surfaces are sealed with silica gel. Modified M-SBF (modified-simulated body fluid) is selected as the corrosive medium. SBF simulated body fluid is a metastable solution, which is an apatite supersaturated solution containing Ca^2+^ and phosphate ions, and is widely used for the evaluation of bioactive materials. A comparison of SBF simulated body fluid and human plasma ion concentration is shown in Table 4. During the electrochemical test, in order to simulate the temperature of the human body, place the corrosive medium in a constant-temperature water tank with the temperature adjusted to 36.5 °C, and let it stand for 10 min before starting the test. After that, the three electrodes are connected.

## 3. Results

### 3.1. Analysis of Phase Composition of Milled Surface

An X-ray diffractometer is used to analyze the phase composition of the machined surface. It is observed that there is no obvious peak in the scanning ranges of 20–25 and 75–90 on the diffraction diagram. For the convenience of observation and comparison, the scanning ranges of 20–25 and 75–90 on the diffraction diagram are removed, and four groups of data are presented in the same diffraction analysis map, as shown in Figure 5.

It can be seen from the observation that the machined surfaces under different cooling conditions are mainly an α-Mg phase and some second phases. After different cooling methods, no new phases appear on the surfaces. The crystallinity of the second phases on the dry and low-temperature machined surfaces is better than that on other machined surfaces, and the number of second phases on the low-temperature machined surfaces is higher than that on other surfaces. The analysis is that liquid nitrogen is sprayed in the cutting area as cutting fluid. A good cooling effect was achieved, and the temperature in the contact area of the chips was reduced. The contents of Ca and Zn exceeded the maximum solubility due to the temperature reduction, and the precipitated Ca and Zn formed the second phase with the Mg matrix. According to Liu’s research [31], with the increase in the second phase, when a large number of second phases are used as cathode phase and magnesium matrix phase to form a corrosive microbattery, the dissolution rate of the magnesium matrix phase is obviously accelerated. However, when the content of the second phase continues to increase to the extent that it can be connected to form a network distribution, and the fine magnesium matrix phase is included in the network second phase at this time, the corrosion propagation rate of the magnesium matrix phase will be significantly reduced, and the corrosion products will also inhibit the continuous corrosion of the matrix phase crystals, thus improving the corrosion resistance of the alloy. Therefore, cryogenic milling contributes to the formation of the second phase, and when enough second phase is formed, it can play a corrosion-resistant role.

### 3.2. Analysis of Tensile Test Results

#### 3.2.1. Elongation and Tensile Strength

The test pieces after tensile fracture is shown in Figure 6 below, and it can be observed that the fracture surface is relatively even, with no obvious necking phenomenon, which accords with the brittle fracture characteristics. In order to observe the changing trend of the six groups of curves, six groups of experimental data are processed into a stress–strain curve, as shown in Figure 7. It is found that the material has no obvious yield stage by observing the curve, and the elongation and tensile strength of the material are calculated, as shown in Table 5. It can be seen that the as-cast magnesium alloy material has the highest elongation at room temperature, which is 2.8%. With the decrease in temperature, the tensile strength increases. At −196 °C, the tensile strength reaches 78.89 MPa, which is 10.68 MPa higher than that at room temperature, while the elongation decreases to 2.11%. It can be determined that the as-cast magnesium alloy material is brittle, and the brittleness of the material increases with the decrease in temperature. Furthermore, by referring to Mahbod [32] and others’ research, it is found that the addition of zinc and calcium will refine the grain size and improve the tensile strength of the material itself.

#### 3.2.2. Tensile Fracture Morphology

Take off the fracture of tensile test pieces and observe its fracture morphology. By observing the electron microscope with a magnification of 500 times, it is found that there are a lot of cleavage steps and quasi-cleavage planes at the fracture. In the scanning electron microscope (SEM) images at 27 °C and 0 °C, dimples were observed, but no dimples were found at other temperatures, as shown in Figure 8 and Figure 9. The appearance of dimples makes the material have better plasticity and higher elongation [33], which can also explain why the elongation of the material decreases with the decrease in temperature.

The magnification was increased 4000 times, and the shallow dimples were found under the electron microscope at −40 °C and −80 °C. The dimples were almost surrounded by cleavage steps and quasi-cleavage planes. With the decrease in temperature, the dimples became shallower and the dimple area became smaller. No dimples were found at −120 °C and −196 °C, which were almost quasi-understanding planes.

With the decrease in tensile temperature, the number of dimples decreases, and the dimple area becomes smaller and smaller. Some crystals have small planes stretching in different directions, which are cleavage planes formed by cracks propagating along different twin planes. Cleavage cracks generally occur on the most densely packed plane of atoms. Cleavage cracks meet with dislocations to form cleavage steps during their propagation, and their direction is usually the same as the crack propagation direction. A large number of such cracks continue to expand and connect, which eventually leads to the fracture failure of the alloy. At low temperatures, such as −120 °C and −196 °C, there are basically no dimples in the fracture surface, but a large number of cleavage planes appear. The appearance of these cleavage planes speeds up the crack propagation in the crystal, and the fracture mode has shown a cleavage fracture mechanism at this time. Cleavage planes in different directions are still connected with each other by tearing ridges extending in all directions. Because the orientation of each crystal is different, the cracks propagate along different planes, and many cleavage steps with different sizes are formed at the intersection of each cleavage plane. With the further aggravation of deformation, the cracks rapidly propagate along these cleavage planes and lead to the final fracture, which shows that the fracture mode of magnesium alloy is a quasi-cleavage fracture, which is consistent with the research conclusion of Guo [34]. It can be seen from the above-mentioned SEM images of fracture morphology that with the decrease in temperature, the dimple area decreases, which indicates that the brittleness of magnesium alloy increases with the decrease in temperature, which explains why the elongation of magnesium alloy decreases with the decrease in temperature in the low-temperature tensile test.

#### 3.2.3. Metallographic Analysis of Tensile Section

Figure 10 shows a metallographic diagram of the tensile section of magnesium alloy with a magnification of 100 times. Through the metallographic diagram, it can be observed that many crystals are separated from the matrix at the fracture, which shows that the fracture mode of the tensile test pieces surface is an intergranular fracture. Intergranular fracture refers to a kind of fracture caused by crack propagation along a crystal boundary in metallic materials. When the fracture surface of the intergranular fracture is granular, it is also called intergranular crystal fracture [35]. In most cases, an intergranular fracture belongs to a brittle fracture. When metal or alloy precipitates continuous or discontinuous network brittle phases along the crystal boundary, under the action of external force, these network brittle phases will directly bear the load, and it is easy to break and form cracks, which will cause the sample to break along the crystal boundary. It is a complete brittle normal break [36]. In the cryogenic milling process, the continuous injection of liquid nitrogen into the cutting area will cause the temperature of the workpiece to be too low. According to the above tensile test, the brittleness of magnesium alloy increases with the decrease in temperature, which is very beneficial to chip breaking, surface quality improvement, and tool life extension. 

### 3.3. Service Performance of Milling Surface

#### 3.3.1. Wear Resistance Characteristics

Because oil-based cutting fluid and water-based cutting fluid are not conducive to the formation of the second phase and cannot have the effect of corrosion resistance, the friction and wear test only compares dry cutting and cryogenic milling. During the friction and wear test, the sensor will measure the friction coefficient between the counter-grinding ball and the sample in real time, and the friction and wear process can be divided into running-in stage and stable wear stage. After the friction and wear test, the change trend of the friction coefficient with time can be derived by computer processing, as shown in Figure 11. The average friction coefficient of the milled surface can be calculated by the software equipped with the friction and wear testing machine, and then the wear morphology of the milled surface can be observed by the white light interferometer, and the wear volume can be calculated. The average friction coefficient and wear volume are shown in Table 6, and the wear morphology of the dry cutting and cryogenic milling surfaces is shown in Figure 12. The displayed value is the relative height value from the lowest wear position to the milled surface, that is, the wear depth value. Fr om the analysis results in Table 6, it can be seen that the average friction coefficient of the low-temperature machined surface is lower than that of the dry-cut machined surface, and the amount of wear is smaller. The analysis reason is that the roughness of the dry-cut machined surface is large, and the residual area of the machined surface increases, resulting in an uneven machined surface, which increases the friction coefficient. In the cooling environment of liquid nitrogen, because liquid nitrogen plays the role of lubrication and cooling, the dislocation slip of crystals becomes difficult at low temperature, resulting in high hardness of the cryogenic milling surface. In the process of friction and wear, the average friction coefficient decreases by about 7.4% compared with dry cutting, and the wear amount decreases by about 10%. From Figure 12, it can be observed that the wear depth of the dry milled surface is 0.14 mm, that of the cryogenic milling surface is 0.12 mm, and that of the wear depth of the dry milled surface is larger, indicating that the wear resistance of the dry milled surface is worse than that of the cryogenic milling surface, which is consistent with the above conclusion of analyzing the average friction coefficient and wear volume. The average friction coefficient of the cryogenic milling surface is lower, the wear amount is smaller, and the wear resistance is better. There are two main reasons: First, because the temperature decreases in the liquid nitrogen cooling environment, the dislocation slip of grains becomes difficult, and the surface hardness increases. Second, the addition of zinc and calcium can improve the hardness of the material after grain refinement [37].

#### 3.3.2. Corrosion Resistance

Similar to the friction and wear test, the corrosion test only compares dry cutting and cryogenic milling, and the microscopic photos before electrochemistry are shown in Figure 13. Using a handheld microscope to observe the corroded surface, it can be seen that after electrochemical corrosion, the surface of the cutting surface turns black from white metallic luster, as shown in Figure 14. According to the first section of the third chapter, it can be known that the second phase appears in magnesium alloy, and the formed second phase can be used as cathode point-to-point accelerated corrosion [38], causing galvanic corrosion of the matrix, so pitting pits mainly appear around the grain boundary of the second phase. The magnesium matrix combines with chloride ions in a corrosive medium to form soluble magnesium chloride, which is fixed on the surface of the alloy matrix to form a circular pit. With the prolongation of the corrosion time, the pit becomes bigger and bigger, and at this time, galvanic corrosion becomes local corrosion. With the prolongation of the corrosion time, the second-phase compound will gradually fall off and lose its protective effect on the alloy matrix.

In the process of the electrochemical corrosion test, the polarization curve can be observed in real time by the equipped software, and the corresponding test data can be recorded. After the electrochemical test, the self-corrosion potential can be checked by the software. The quotient of corrosion current and corrosion surface area is corrosion current density, and the corrosion resistance of the milled surface can be judged according to the corrosion current density and self-corrosion potential. The polarization curves of the dry milled surface and cryogenic milling surface obtained by processing the test data are shown in Figure 15. Table 7 shows the self-corrosion potential and corrosion current density of machined surfaces under different cooling conditions. As shown in the figure, compared with the surface machined by dry cutting, the surface machined by cryogenic milling has higher self-corrosion potential and lower corrosion current density, indicating that the corrosion resistance of the surface machined by cryogenic milling is better than that of the surface machined by dry cutting. Combined with the research results in Chapter 3, it is analyzed that the poor corrosion resistance of the dry milled surface is due to the high roughness of the dry milled surface and more metal residues on the surface, resulting in an uneven distribution of the oxide film formed by machining. In this case, corrosion reaction is more likely to occur, so the self-corrosion potential of the dry milled surface is low, and the corrosion current density is high. However, the surface roughness of cryogenic milling is low, the oxide film formed is relatively dense and uniform, and the refinement degree of the surface crystal after cutting is better than that of dry cutting, so the corrosion resistance of the cryogenic milling surface is better. There are two main reasons: First, with the increase in the content of the second phase, when it can be connected to form a network distribution, the fine magnesium matrix phase is included in the network second phase at this time, so the corrosion propagation rate of the magnesium matrix phase will be significantly reduced. Second, because corrosion products can also inhibit the continuous corrosion of matrix phase grains, the surface roughness of low-temperature cutting is low, the oxide film formed is relatively dense and uniform, and the refinement degree of surface grains after cutting is better than that during dry cutting, so the corrosion resistance of the low-temperature cutting surface is better.

## 4. Conclusions

In this paper, cutting, the tensile test, and the surface service performance test of the prepared magnesium alloy show that: Under different cooling conditions, XRD analysis found that the solubility of Ca and Zn in the Mg matrix decreased due to the low-temperature environment caused by liquid nitrogen injection in the cutting area, which promoted the formation of more second phases on the machined surface. When enough second phases were formed, the corrosion propagation rate of the Mg matrix phase decreased significantly, and the corrosion products also inhibited the continuous corrosion of matrix phase crystals, thus improving the corrosion resistance of the alloy.With the decrease in material temperature, the tensile strength of magnesium alloy increases by 10.68 MPa, and the elongation decreases to 2.11%. The internal tensile fracture mode of the material is a quasi-cleavage fracture, and the surface of the material is an intergranular fracture. With the decrease in temperature, the brittleness will be further enhanced, which will promote the surface quality.Through the friction and wear test, it is found that in the environment of liquid nitrogen cooling, because liquid nitrogen plays a role in lubrication and cooling, the dislocation slip of crystals becomes difficult, resulting in the high hardness of the cryogenic milling surface. Compared with the dry milled surface, the average friction coefficient is about 7.4% lower, and the wear amount is about 10% lower. Through the electrochemical corrosion test, it is found that the surface roughness of cryogenic milling is low, the oxide film formed is relatively dense and uniform, and the refinement degree of the surface crystal after cutting is better than that of dry cutting, so the corrosion resistance of the cryogenic milling surface is better.

## Figures and Tables

**Figure 1 jfb-13-00179-f001:**
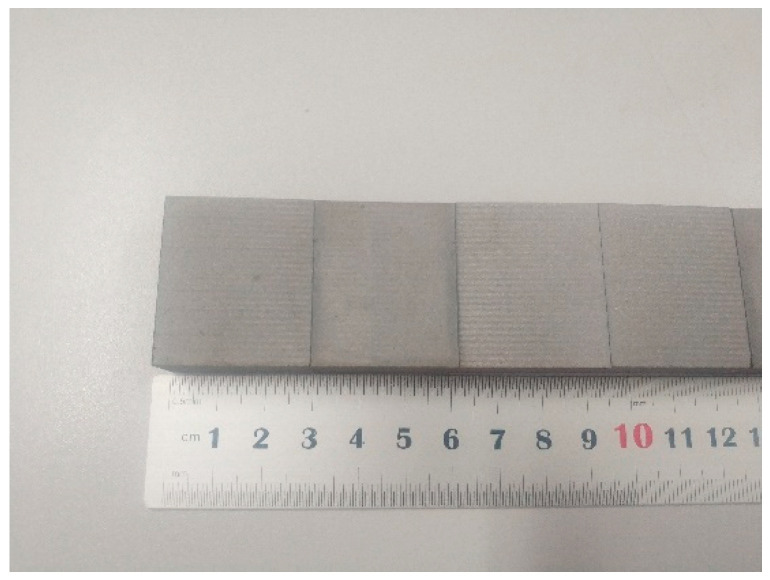
Magnesium alloy after cutting.

**Figure 2 jfb-13-00179-f002:**
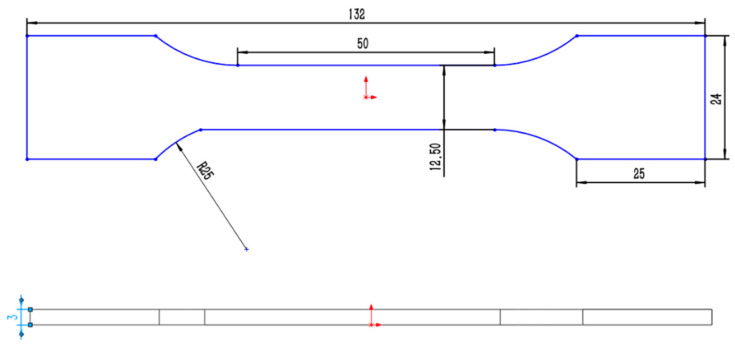
Drawing of tensile test pieces.

**Figure 3 jfb-13-00179-f003:**
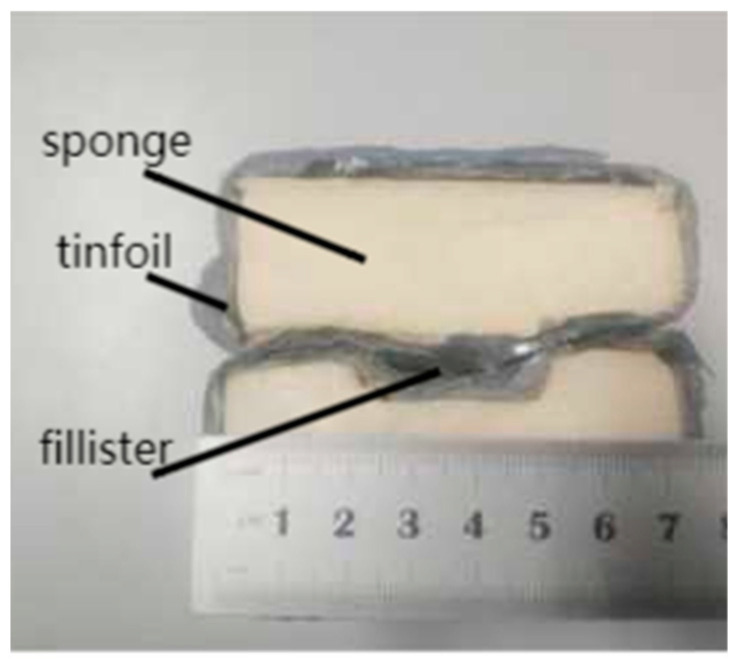
Physical drawing of the thermal insulation device.

**Figure 4 jfb-13-00179-f004:**
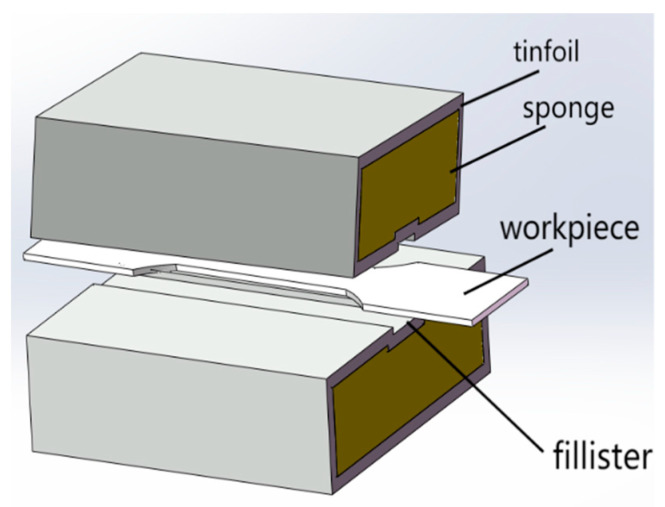
Schematic diagram of the relative position between test pieces and the thermal insulation device.

**Figure 5 jfb-13-00179-f005:**
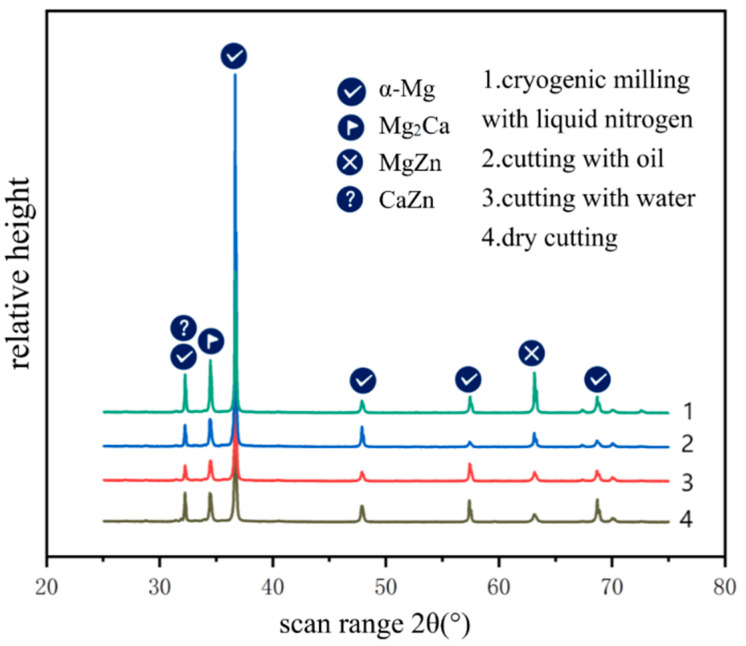
Diffraction analysis map of magnesium alloy.

**Figure 6 jfb-13-00179-f006:**
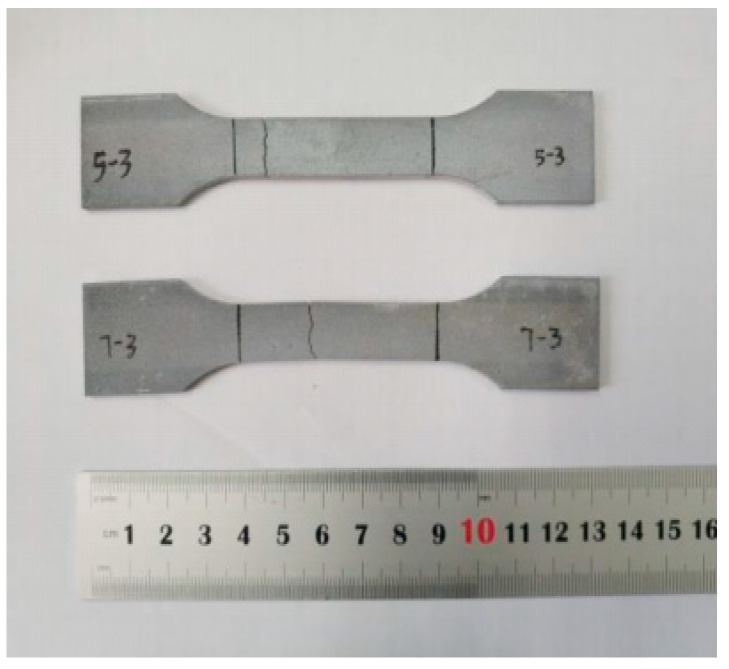
Magnesium alloy fractured by tension.

**Figure 7 jfb-13-00179-f007:**
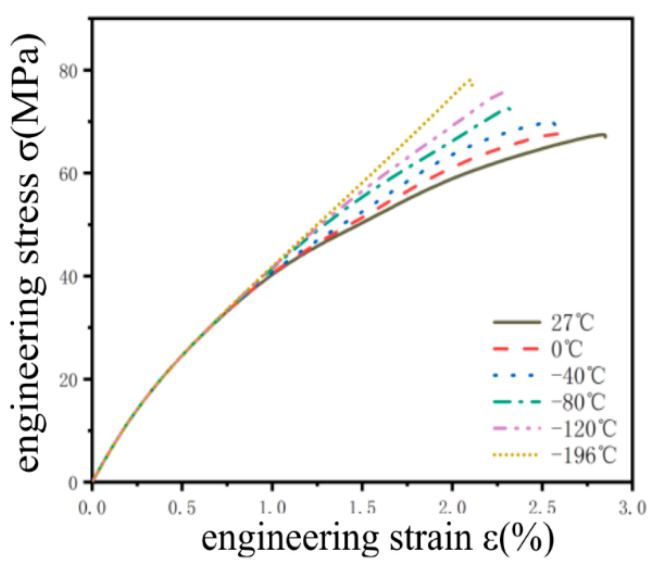
Stress–strain curve of magnesium alloy.

**Figure 8 jfb-13-00179-f008:**
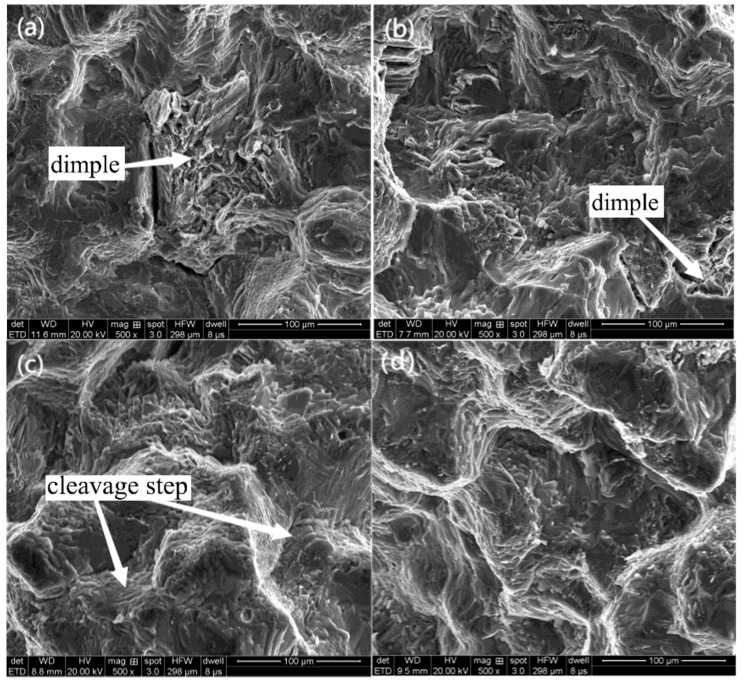
Scanning electron microscopy of the tensile fracture of magnesium alloy (500×): (**a**) 27 °C, (**b**) 0 °C, (**c**) −40 °C, (**d**) −80 °C, (**e**) −120 °C, (**f**) −196 °C.

**Figure 9 jfb-13-00179-f009:**
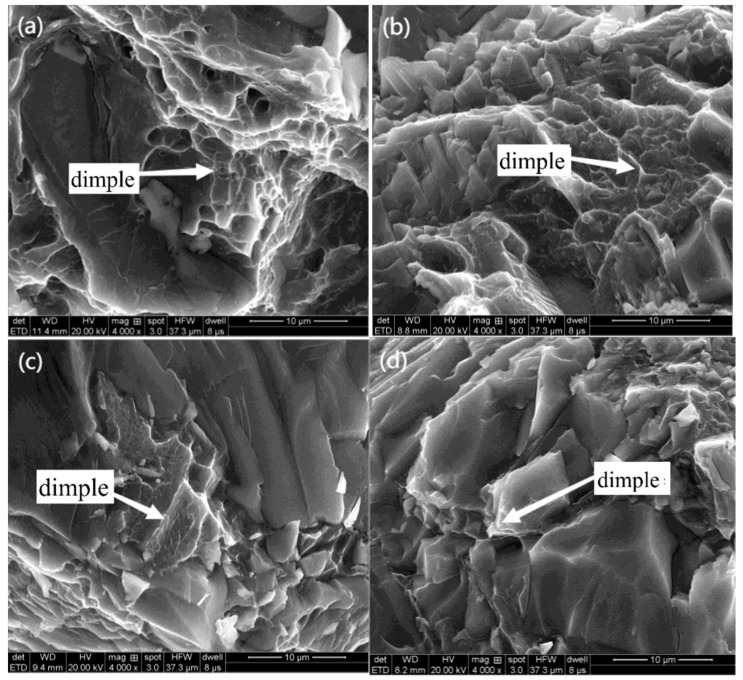
Scanning electron microscopy of the tensile fracture of magnesium alloy (4000×): (**a**) 27 °C, (**b**) 0 °C, (**c**) −40 °C, (**d**) −80 °C, (**e**) −120 °C, (**f**) −196 °C.

**Figure 10 jfb-13-00179-f010:**
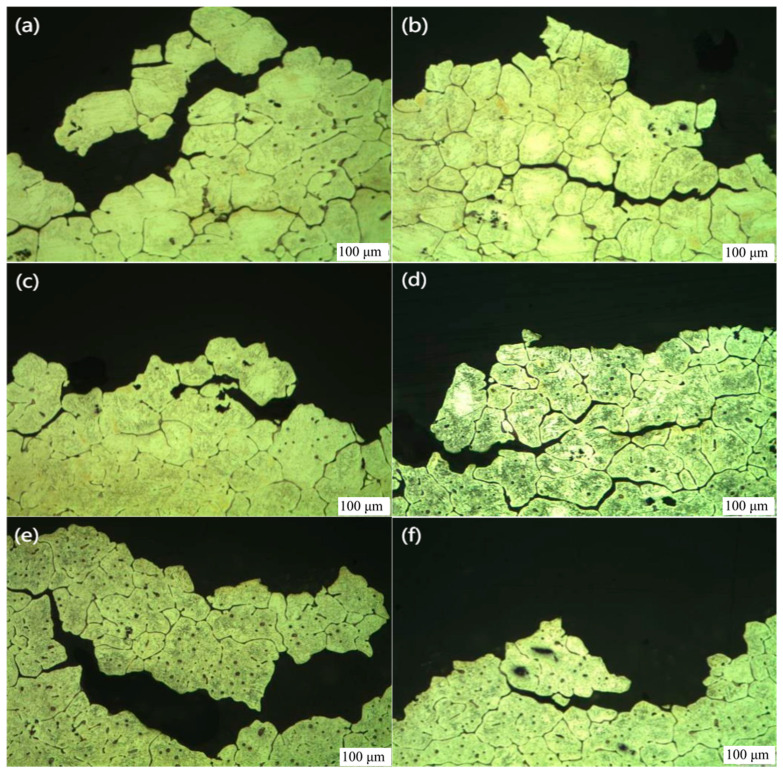
Metallographic diagram of the tensile section of magnesium alloy (100×): (**a**) 27 °C, (**b**) 0 °C, (**c**) −40 °C, (**d**) −80 °C, (**e**) −120 °C, (**f**) −196 °C.

**Figure 11 jfb-13-00179-f011:**
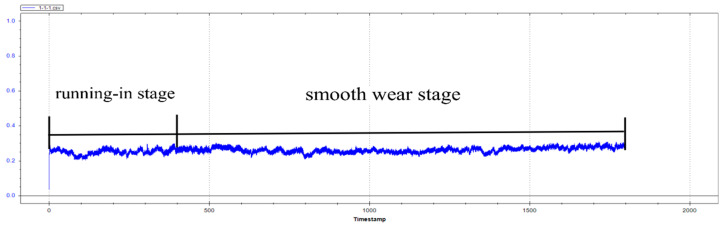
Trend of the friction coefficient over time.

**Figure 12 jfb-13-00179-f012:**
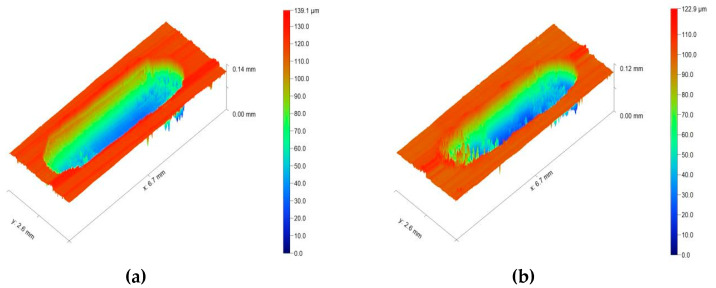
(**a**) Surface wear morphology of dry cutting. (**b**) Surface wear morphology of cryogenic milling.

**Figure 13 jfb-13-00179-f013:**
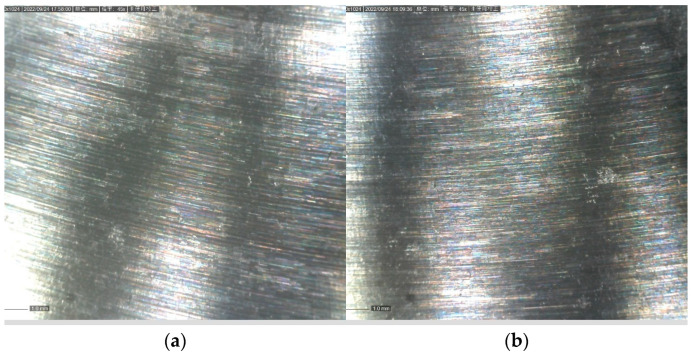
(**a**) Electrochemical milling of the surface at normal temperature before. (**b**) Electrochemical low-temperature milling of the surface before.

**Figure 14 jfb-13-00179-f014:**
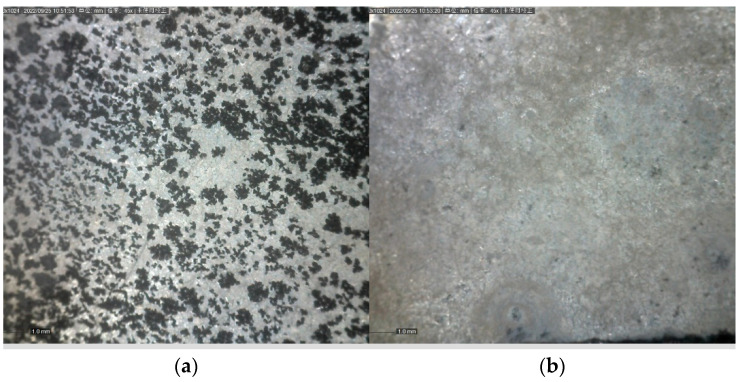
(**a**) Electrochemical milling of the surface at normal temperature after. (**b**) Electrochemical low-temperature milling of the surface after.

**Figure 15 jfb-13-00179-f015:**
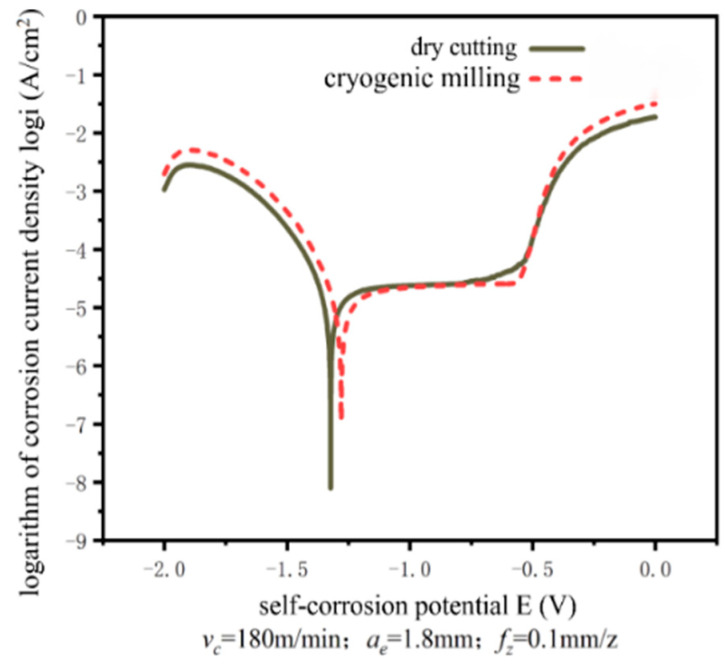
Polarization curve of the milled surface.

**Table 1 jfb-13-00179-t001:** EDS analysis results of magnesium alloy.

Element	Element Concentration	Intensity Correction	Weight Percentage	Atomic Percentage
Mg	47.37	1.3069	96.24	98.19
Ca	0.56	0.9379	1.61	1.00
Zn	0.65	0.8203	2.15	0.81

**Table 2 jfb-13-00179-t002:** Configuration parameters of the YCM-V16b vertical machining center.

Parameter	Maximum	*X*-Axis Stroke	*Y*-Axis Stroke	*Z*-Axis Stroke	Taper of Spindle End Hole
Parameter	6000	1100	600	630	BT50

**Table 3 jfb-13-00179-t003:** Parameters of the keyway milling cutter.

Product Model	Material Quality	Coating	Hardness	Blade Diameter	Blade Length	Overall Length	Tooth Numbers
GM-2E	Tungsten steel	TIAIN	HRC50°	20	45	100	2

**Table 4 jfb-13-00179-t004:** Comparison of simulated body fluid SBF and human plasma ion concentration.

Component	Na^+^	K^+^	Mg^2+^	Ca^2+^	Cl^−^	HCO^3−^	HPO_3_^2−^	SO_4_^2−^
SBF solution	142.0	5.0	1.5	2.5	103.0	10.0	1.0	0.5
Plasma	142.0	5.0	1.5	2.5	103.0	27.0	1.0	0.5

**Table 5 jfb-13-00179-t005:** Elongation and tensile strength of magnesium alloy.

Temperature	27 °C	0 °C	−40 °C	−80 °C	−120 °C	−196 °C
Tensile strength (MPa)	68.21	69.1	70.7	73.4	76.38	78.89
Elongation (%)	2.8	2.65	2.57	2.32	2.28	2.11

**Table 6 jfb-13-00179-t006:** Average friction coefficient and wear volume.

Cutting Way	Average Friction Coefficient	Wear Volume
Dry cutting	0.2815	1.54175
Cryogenic milling	0.2607	1.38562

**Table 7 jfb-13-00179-t007:** Self-corrosion potential and corrosion current density of the machined surface.

Cutting Way	Self-Corrosion Potential E (V)	Corrosion Current Density I (A/cm^2^)
Dry cutting	−1.323	1.3 × 10^−5^
Cryogenic milling	−1.28	1.042 × 10^−5^

## Data Availability

Not applicable.

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
