# Peer review of "Characteristics and Surface Serviceability for Cryogenic Milling Mg-1.6Ca-2.0Zn Medical Magnesium Alloy"

_jfb, 2022, doi:10.3390/jfb13040179_

Round 1
Reviewer 1 Report
Dear autors
Professional articles should contain detailed information on the test results, not photos of basic equipment such as a milling machine or samples for testing. In addition, the article is full of bugs. How is it possible that in the Mg-1.6% Ca-2.0% Zn alloy, the mass fraction of Ca is 96% (table 1)? What is "texure of wood - Tungsten steel" in table no. 3? The article needs to be re-edited
Reviewer 2 Report
This work is of great interest. The work presents the surface properties of Mg-Ca-Zn after cryogenic milling under different cooling conditions. The microstructure and tensile properties were presented in the paper. The article is informative and suitable for publication, but the following changes need to be made to improve the article.
1. Some sentences are very long, e.g., sentences belong to the line numbers 36-40, 46-51, and 60-65. It is suggested that they be split into shorter sentences.
2. Figure 18 has two panels. The panels should be listed as (a) and (b).
Reviewer 3 Report
General comment
This work deals with the characterization of machined surfaces of the Mg-1.6Ca-2.0Zn alloy. The topic is certainly interesting but the manuscript in the present form should not be accepted for publication. In fact, it is not well-written, English language should be significantly improved, the meaning of several sentences is not clear, and the information are not properly organized. In particular, introduction should be revised and the advancement with respect to the state of the art and the objective of this investigation should be clearly indicated and highlighted.
Other comments:
1. Several typos are present throughout the text;
2. The authors used the term “cutting” in the case of dry machining, while milling is adopted when coolants are used. This choice it is unclear to me.
3. Some acronyms are reported without explanation;
4. Figures 1, 3, 5, 9, 10 are not needed and they can be deleted;
5. Several statements or conclusions are reported without the corresponding experimental results or references For instance, see may last comment;
6. Figures 7 and 8 are unclear. They should be combined and a detailed description should be reported in the figure caption.
7. In the experimental section, the adopted procedure to produce the alloy should be reported in detail.
8. Many parts of the experimental section are written as a “cooking recipe”: “put this”…, “do that…”. This is strongly unusual and not appropriate for a scientific paper. Please, revised.
9. Figure 19; sample surface should be reported before and after electrochemical test for both dry and cryogenic cut samples.
. Page 13, lines 327-329. The sentence should be confirmed by showing the related imagine.
Round 2
Reviewer 1 Report
Overall, the article is at an average level. Too many cited articles (13) in Chinese. No possibility to verify the correctness of citations. For example, the text in lines 42 and 43 refers only to articles in Chinese "..... and cryogenic milling with liquid nitrogen can not only avoid corrosion, but also meet the requirements of" green manufacturing "[19-22] ,. .... ". Moreover, lack of scrupulousness. The hardness HRC of the tool in table 3 is given in degrees "HRC50o"? In addition, what is "thungsten steel"? Meaby it is high speed steel with thungsten? Despite this the article can be published
Reviewer 3 Report
The manuscript in the revised form can be accepted for publication.